# Plant Gravitropism and Signal Conversion under a Stress Environment of Altered Gravity

**DOI:** 10.3390/ijms222111723

**Published:** 2021-10-29

**Authors:** Dan Qiu, Yongfei Jian, Yuanxun Zhang, Gengxin Xie

**Affiliations:** 1Center of Space Exploration, Ministry of Education, Chongqing University, Chongqing 400044, China; 201826021016@cqu.edu.cn (Y.J.); yuanxun.zh@cqu.edu.cn (Y.Z.); 2Chongqing Key Laboratory of Biology and Genetic Breeding for Tuber and Root Crops, School of Life Sciences, Chongqing University, Chongqing 401331, China

**Keywords:** plant gravitropism, signal conversion, abiotic stress, altered gravity

## Abstract

Humans have been committed to space exploration and to find the next planet suitable for human survival. The construction of an ecosystem that adapts to the long-term survival of human beings in space stations or other planets would be the first step. The space plant cultivation system is the key component of an ecosystem, which will produce food, fiber, edible oil and oxygen for future space inhabitants. Many plant experiments have been carried out under a stimulated or real environment of altered gravity, including at microgravity (0 g), Moon gravity (0.17 g) and Mars gravity (0.38 g). How plants sense gravity and change under stress environment of altered gravity were summarized in this review. However, many challenges remain regarding human missions to the Moon or Mars. Our group conducted the first plant experiment under real Moon gravity (0.17 g) in 2019. One of the cotton seeds successfully germinated and produced a green seedling, which represents the first green leaf produced by mankind on the Moon.

## 1. Introduction

Earth’s life adapts to environmental factors (temperature, humidity, gravity etc.) and is very successful at survival [1,2,3]. Plants exhibit strong life adaptability during a single environmental change or multiple environmental changes [4]. The scientific significance of plant microgravity experiments in space biology is to study the mechanism of constructing spatial biological regeneration systems and plant sensing gravity. An important part of human space exploration is the construction of a plant-based life regeneration ecosystem [5].

Plant biology experiments in ground-controlled environments have contributed to simulate the various changes in plant growth under a space environment of altered gravity [6,7]. With the establishment of the International Space Station (ISS), plant experiments under real space microgravity have become possible. With the development of the space agencies of increasing countries, the experimental facilities in the ISS have been upgraded and improved. In recent years, basic research on plant gravity response and signal conduction has been widely developed. It is believed that adaption of roots and stems growth of plants to the microgravity conditions will be essential [7].

However, the development of human space science can not only be carried out in space stations but must also strive to find the next planet suitable for human habitation. Many studies have been carried out to simulate the low-gravity environment of planets such as the Moon and Mars [7,8,9,10]. With the successful landing of the Chang’e 4 probe at the far side of moon, our group conducted a scientific experiment on plant growth. The cotton seeds were germinated and grown as the first-ever plant on the Moon.

The goal of the human space exploration is to further explore outer space (Mars, the Moon etc.) [11]. The construction of a life-regeneration ecosystem on other planets is an important technical issue in space biology [12]. Therefore, the simulation of space plant growth experiments and the development of planetary surface plant experiments must be further emphasized [13].

## 2. History of Plant Cultivation Systems in Outer Space

Food and oxygen production, carbon dioxide removal, waste recycling and water purification by a micro-life support system (MLSS) are some of the key technologies and problems for long-term manned interstellar travel and immigration. Many studies have focused on screening a large number of organisms suitable for growth in harsh space environments and studying how they can adapt to the extreme environment of outer space. The space plant cultivation system is the key component of an MLSS, and space plants have been cultivated on four major research platforms and stages [12,14]. In early 2014, NASA launched the “Vegetable Production System (VEGGIE)”, which is the first system designed for food production under microgravity, and as a result, astronauts ate self-grown lettuce on the ISS [14].

In space plant cultivation systems, early-stage tasks include developing small-scale plant growth systems to produce fresh vegetables and small fruits to supplement astronaut diets. In the later stages, as the duration of the task increases, plants will provide an increasing number of important functions, such as food and oxygen production, carbon dioxide removal and water purification, all of which require innovative horticultural techniques and methods [15,16,17]. These technologies will include the development of efficient electrical lighting or solar collectors; innovative design and management of greenhouses and planting modules; and the development of recycling technology to preserve water and nutrients [18,19]. Therefore, determining how to successfully cultivate plants in space will require screening a large number of species and meeting the functional needs of the above mentioned modules [12].

The ISS has been served by joint efforts of the aerospace industry across various countries [20,21]. The space station laboratory has evolved from the most basic simple plant growth carrier to a complete plant test system with high-end molecular facility instruments, which from simple observations of plant growth to complex biological component determinations and the detection of changes regarding the plant molecular levels [22]. The space plant cultivation system changed from the Oasis series in 1970 to the VIEGGIE launched by NASA in early 2014 and has been optimized for plant growth space, light supply and cultivation environment [15,16,23]. The space station plant culture room, as the core carrier, provides an experimental guarantee for the feasibility and accuracy of plant microgravity studies [24].

## 3. How Plants Sense Gravity and Change under Stress Environments of Altered Gravity

Plants evolve and grow under the influence of constant factor, Earth’s gravity (1 g). In response to this pressure, plants have acquired gravitropism to sense gravity and change their growth direction and morphogenesis [25,26]. To change the intensity of gravity of ground simulation to study plant changes is a common method to explore the mechanism of plant gravitropism [4,27]. Through long-term research, the molecular mechanisms for sensing gravity were disclosed in different plant species, especially in the model organism *Arabidopsis thaliana* [28,29]. *Arabidopsis* has the advantages of small genome, short life cycle, easy planting, prolific seed, and a large number of mutants compared to other plant species [30,31].

### 3.1. How Plant Sensing Gravity

Gravity sensing in plants is a very complex process, including the perception of gravity by cells and signal transduction in cells as well as the cells’ responses to change [32]. Amyloplast re-precipitation in root columella cells is a critical initial step in gravity sensing when the plant roots are laterally reoriented. This process somehow causes cytoplasmic alkalization of these cells and then repositions the auxin efflux vector (PIN genes) [33]. This changes the auxin flow throughout the root, producing a lateral gradient of auxin throughout the cap that causes different cell elongation and gravity when delivered to the elongated region. Recent studies showed the evidence that these participants transferred signals from amyloplasts deposits to the auxin signaling transduction: mechanically sensitive ion channels, actin, calcium ions, inositol triphosphates, receptors/ Ligand, ARG1/ARL2, spermine and TOC complex [34,35].

The elongation zone is also a key part of root bending during gravity sensing. Gravitropism is the result of different accumulations of auxin on either side of the elongation zone, resulting in differential growth curvature [36]. Many experiments have demonstrated the importance of precipitating amyloplasts in gravitropism, and the precipitation of dense amyloplasts is a critical first step [36,37,38,39]. Amyloplast-free mutants still responded to gravity, and some studies revealed the possible existence of another gravity-sensing mechanism [39]. In addition to the root cap, there is another gravity sensing site that is dependent on actin, which is different from the mechanism of columnar actin. Some plant cells can also sense gravity by the hydrostatic pressure exerted by the protoplasts on the cell wall [38].

### 3.2. Transduction of Gravity Signal and Auxin as a Signal

After the plants turned to other orientations, the amyloplasts of the root columella cells began to drop to the cell’s new bottom-side [35,40]. Auxin is transmitted by columella cells to the elongation zone to initiate the signal transduction [41]. The auxin influx vector AUX1 is required for the transmission of gravity signals, and AUX1 plays a crucial role in the downstream step of the auxin gradient from the root to the elongation zone.

The auxin efflux vector dynamically controls the flow of auxin during gravity; conversely, the PIN protein releases auxin from the cell, and PIN2 is critical for the transport of this differential fluid through the lateral cover and upward through the extension band [42]. The Ca^2+^ signal is located downstream of the auxin signal and is involved in the conversion to the auxin signal to the change in extracellular pH value [43]. Inositol triphosphate may be involved in the interaction between light and gravity reactions. The previous research suggested that the inositol triphosphate signaling pathway may be involved in the repositioning of the PIN protein by regulating the intimal system after root reorientation in the gravitational field. Whether this process depends on Ca^2+^ remains to be elucidated [44].

The asymmetric distribution of auxin requires the auxin transporters AUX1 and PIN2 [36]. However, the relationship between AUX1 and PIN2 is unclear [44]. There is a report that demonstrated that the *aux1-T* mutant exhibits a stronger defect in root gravity than *pin2-T* and that the *aux1-T/pin2-T* double mutant exhibits an agravitropic phenotype similar to *aux1-T*. In the *pin2-T*, *aux1-T* and *aux1-T/pin2-T* mutants, the gravity-induced auxin response asymmetric distribution could not be established; whereas the *aux1-T/pin2-T* double mutant responded the same way as the *aux1-T* mutant. These results support the role of AUX1 in upstream of PIN2 [45].

### 3.3. The Dynamic Model of Amyloplast Sedimentation

The dynamic model of amyloplast sedimentation of the root columella cells is calculated using the following equation. From the differential equation of the motion of a substance in a liquid, the following can be obtained:
$$\frac{mdv}{dt}=mg-F_{b}-kv$$

In the formula, *m* is the mass of the amyloplasts; *F_b_* is the buoyancy of the amyloplasts in the cytoplasm; *k* is a constant related to the viscosity of the cytoplasm, which depends on cell volume; and *v* is the speed of amyloplast movement (Figure 1).

When the initial condition *t* = 0 is set and *v* = 0, *v* can be solved by solving the differential equation.
(1)$$v=\frac{mg-F_{b}}{k}(1-e^{-\frac{kt}{m}})$$

According to Newton’s second law, fluid dynamics and other theories, the resultant force of the amyloplasts in the cytoplasm can be obtained with the following equation:(2)$$F=G-F_{b}-F_{r}=\rho_{1}Vg-\rho_{2}Vg-kv$$

In Formula (2), *F* is the combined force shown by the amyloplasts, *G* is the gravity of the amyloplasts, *F_b_* is the buoyancy of the amyloplasts, *F_r_* is the viscous resistance of the amyloplasts when they move in the cytoplasm, *ρ*_1_ is the amyloplast density, *ρ*_2_ is the density of the cytoplasm, and *V* is the volume of the amyloplasts (Figure 1).

Then,
(3)$$a=\frac{F}{\rho_{1}V}=\frac{\rho_{1}-\rho_{2}}{\rho_{1}}g-\frac{kv}{\rho_{1}V}=\frac{\rho_{1}-\rho_{2}}{\rho_{1}}ge^{-\frac{kt}{\rho_{1}V}}$$
(4)$$s=\frac{1}{2}at^{2}=\frac{1}{2}\frac{\rho_{1}-\rho_{2}}{\rho_{1}}e^{-\frac{kt}{\rho_{1}V}}gt^{2}$$

In the formula, s is the displacement of amyloplasts in the cytoplasm.

Equation (4) simplifies to
(5)$$s=k_{1}e^{-k_{2}t}gt^{2}$$

Thus, $k_{1}=\frac{1}{2}\frac{\rho_{1}-\rho_{2}}{\rho_{1}}$, $k_{2}=-\frac{k}{\rho_{1}V}$

### 3.4. The Lateral Root Sense Altered Gravity

Recently research demonstrated that lateral roots can serve as a good system for exploring amyloplast-dependent mechanisms [46]. *Arabidopsis* lateral roots have stronger amyloplast-dependent gravitational pathways than primary roots. There is evidence that an amyloplast -independent mechanism plays a role in primary roots; however, this is difficult to determine [18].

### 3.5. The Plant Gravitropism Related to Phototropism

Light regulates many physiological processes related to plant development, thus, affecting seed germination and seedling morphology, especially in the activation and regulation of cellular and molecular functions [47,48]. The interaction between light and gravity induction of plants was studied under the action of spatial microgravity [49,50]. In fractional gravity studies, the wild-type and mutant phytochrome A and B genotypes of *Arabidopsis thaliana* showed an attenuation of red-light-based phototropism in both roots and hypocotyls of seedlings occurring due to gravitational accelerations ranging from 0.l to 0.3 g. In contrast, blue-light negative phototropism in roots, which was enhanced in microgravity compared with the 1*g* control, showed a significant attenuation at 0.3 g [51].

## 4. The Plant Experiments under Simulated and Real Altered Gravity Environment

### 4.1. The Plant Experiments under Simulated Microgravity (0 g), Moon Gravity (0.17 g) and Mars Gravity (0.38 g) Environment

Numerous studies on plant growth and development have been conducted in altered gravity environments since the beginning of human spaceflight; however, how gravity affects plant growth and development is still unclear [9,52,53,54]. Inclinometers and Random Positioning Machine (RPM) are used to simulate microgravity. Two types of methods using RPM to simulate partial gravity were developed, one by implementing a centrifuge on the RPM and the other by driving a RPM motor using a specific software protocol [55,56].

Gene expression levels under microgravity conditions involved in regulating the cell polarity, cell wall development, oxygen status, and cell defense or stress were found to be more than twice as large as those under earth gravity experiments, which indicates microgravity as an important factor affecting these key genes [29]. The effects of light and gravity on plants were investigated under microgravity conditions, and the photoreaction of blue and red light in *Arabidopsis* roots was found, indicating an antagonistic relationship between light and gravity signals during early plant growth and development [48,50]. Furthermore, the epigenetic modifications of chromatin and a serious disturbance of cell proliferation were identified due to altered gravity effects that affect the cellular functions for normal plant development [8,36,57].

In addition, the cell proliferation of fixed root meristematic cells from 4-day grown *Arabidopsis thaliana* seedlings appeared increased, and cell growth was depleted under Moon gravity compared with the 1 g control [52]. However, the results at the simulated Mars level were close to the 1 g static control, which suggests that the threshold for sensing and responding to gravity alteration in the root would be at a level intermediate between Moon and Mars gravity [52]. Depending on the organisms examined, studies suggest that the threshold for gravity required for living systems to operate normally is 0.3 g or less, which is supported by other observations [4,9,46,54].

In another study, the in vitro cell culture of *Arabidopsis thaliana* was placed under different conditions of simulated microgravity, at Mars gravity (0.38 g) and hypergravity (2 g), to study the cell proliferation, growth and appearance [8,58]. The most relevant changes occurred in the 24-h treatment, which was more pronounced for simulated gravity decline than for supergravity, which indicated that changes in gravity effects include severe interference with cell proliferation and growth [58]. The key transcripts responded to altered gravity have been identified. For example, 396 transcripts were at least 100% upregulated during the microgravity phase of the parabola, among them 25 Ca^2+^-dependent genes, such as members of the Ca^2+^-binding protein family, or Ca^2+^-dependent protein kinases [59].

### 4.2. The Plant Experiment under a Real Moon Gravity Environment (0.17 g)

Our group conducted the first plant experiment under real Moon gravity (0.17 g) in 2019. The biological experiment payload (BEP) is a cylindrical structure with a weight of 2.608 kg, which meets the requirements of the Chang’e 4 probe. The BEP includes a control module, a thermal control module, a structural module, a light-guiding module, and a biological module. The biological space has a volume of approximately 0.82 L, contains cotton seeds, rapeseed seeds, potato seeds, *Arabidopsis* seeds, insects and yeast, and constitutes a micro-life cycling system. The BEP has a very small light-transmitting hole with a diameter of Φ10.

The BEP was installed on the lander of the Chang’e 4 probe and successfully completed the first-ever soft landing on the far side of the Moon at 10:26 a.m. on 3 January 2019, Beijing time (Figure 2). The ground-controlled BEP was set up at the same time. Surprisingly, a cotton seed in the BEP on the Moon germinated 22 h from the water injection time (WIT). The first leaf of the cotton seedling in the BEP on the Moon was observed at 82 h from the WIT. There was no large difference in leaf size between the two pictures taken at 82 and 190 h, suggesting that growth retardation occurred at this stage. Cotton seeds were germinated in the ground-controlled BEP 53 h from the WIT. The seedlings grew rapidly, and only the stalk was captured by the camera after 190 h (Figure 2). No other sign of germination in the BEP of the Moon except for the cotton seedling was observed within the view of the camera.

The growth retardation of the cotton seedling observed in this study is consistent with previous simulation studies [4,9,46,54]. Plant growth and development is greatly affected by the actual lunar microgravity environment. Under the 1 g gravity circumstance, the amyloplasts in the columella cells will fall to the lower side of the cell membrane in response, which is the key step in gravity sensing [35,46]. The following activating of PIN3 and PIN7 accumulation in the lower side of the cell membrane could cause rapid efflux of auxin [34,45]. In this case, the auxin content in the upper side of the root tip is higher than the auxin content in the lower side, resulting in rapid growth of the cells in the upper part of the root tip. The root will grows in the direction of gravity [39,41,49].

Why amyloplasts fall in gravity sensing? The amyloplasts has a density of 1.5 g/cm^3^, whereas the surrounding cytoplasm has a density of approximately 1.02–1.1 g/cm^3^. Will this falling change according the gravity variation? A dynamics model of amyloplast sedimentation in cytoplasm under gravity variation was set up based on Newton’s second law (Figure 1). The fluid dynamics theories are shown in Figure 3a.

As the gravity declined, the descent velocity of amyloplasts declined in the dynamics model, in which 0.25 g and 0.11 g were the two turning points. This indicates that the displacement of amyloplasts will be much slower under 0.25 g or less gravity circumstances (Figure 3b). This will affect the development of the plant root system and, subsequently, the growth of the whole plant, which made the horizontal elongation of embryo roots in BEP on Moon under only 1/6 g (Figure 3b). This might be the prime cause of the threshold for gravity required for plant living systems to operate normally, which was found to be 0.3 g or less from other studies; however, further investigation and evidence, such as actual microscope captures with amyloplast displacement under 0.3 g gravity, will be needed [4,9,46,54].

## 5. Concluding Remarks and Future Perspectives: Mars and Lunar Base Construction

Over the years, humans have been committed to space exploration and finding the next planet suitable for human survival [60,61]. There are hopes to build an ecosystem that adapts to the long-term survival of human beings in space stations or other planets [61]. Many plant cultivation system have been carried out on the ISS, which are related to the effects of microgravity on plant growth [14,62]. However, many challenges remain regarding human missions to the Moon or Mars. We present a few suggestions that might be helpful.

First, establishing a base, cultivating crops and maintaining a biological life support system (BLSS) on the Moon or Mars will be very difficult. Not only gravity alterations but also magnetic field alterations, photoperiod alterations, extreme environmental temperatures and unforeseeable radiation will be major obstacles to crop cultivation systems on the Moon or Mars [9,52].

Second, more on-ground experiments simulating specific Moon or Mars conditions are required. Fundamental research on higher-plant regulatory networks at the molecular level is urgently needed to understand the molecular signals in the distinct phases of the stress response and adaptation. Such information will speed up the development of crop varieties without gravitropism and magnetoreception but with the ability to quickly adapt to other long-term abiotic stresses [59].

Third, the performance of each organism and the interactions among organisms in the BLSS should be investigated [4,24]. Advanced cultivation facilities exploring the benefits of candidate plant species for a specific function, such as resistance to gravity alterations, a high rate of oxygen production, a high biomass production rate and radiation resistance, are needed.

## Figures and Tables

**Figure 1 ijms-22-11723-f001:**
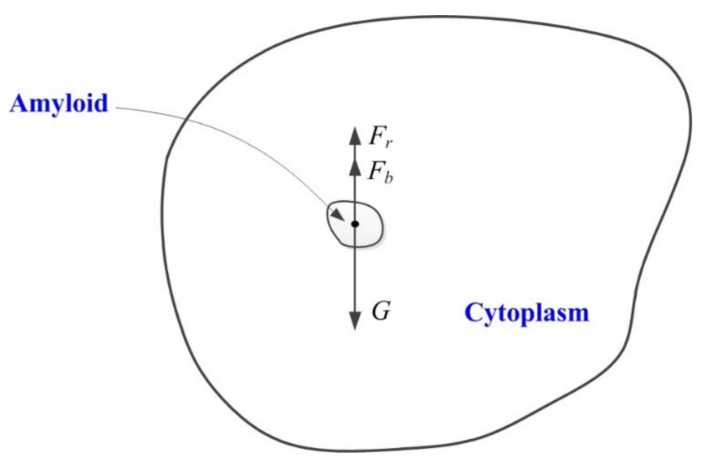
Dynamics model of amyloplasts sedimentation in cytoplasm. The big black irregular circle represents the root columella cell. The small black irregular circle represents the amyloplast (amyloid). *F* is the combined force shown by the amyloplasts, *G* is the gravity of the amyloplasts, *F_b_* is the buoyancy of the amyloplasts, and *F_r_* is the viscous resistance of the amyloplasts when they move in the cytoplasm.

**Figure 2 ijms-22-11723-f002:**
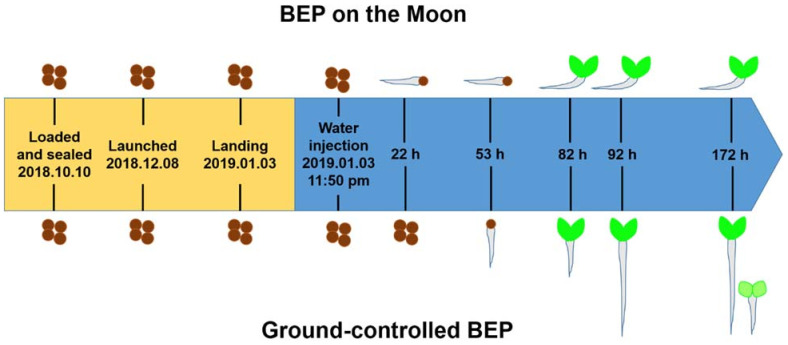
Brief comparison between the ground-controlled BEP and the BEP on the Moon. The brown circles represent seeds. The brown circles with a light grey tail represent germinated seeds. The green seedlings represent the seedlings of cotton or rapeseed. The time from the WIT is indicated in h. The germination of the cotton seed on the Moon was much faster than that on Earth. Horizontal elongation of the embryonic root and growth retardation 92 h after the WIT were observed in the BEP on the Moon.

**Figure 3 ijms-22-11723-f003:**
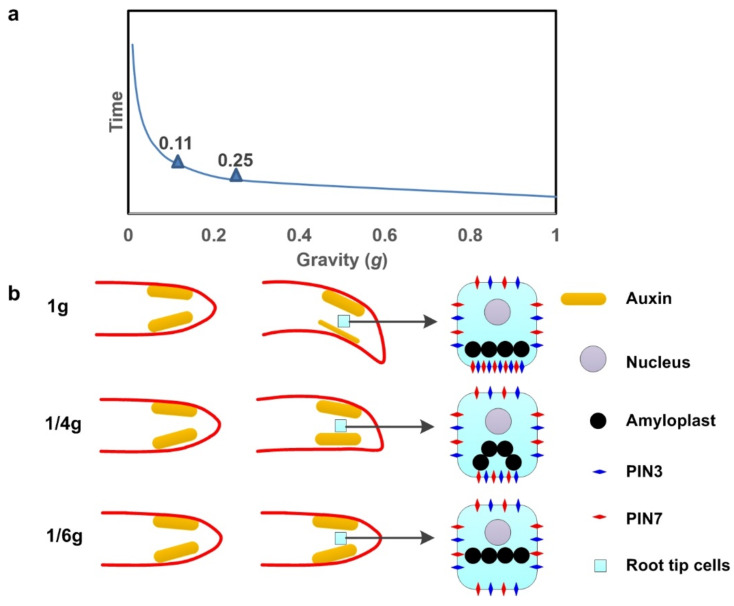
Dynamic model of amyloplast sedimentation in the cytoplasm. (**a**) The correlation between the descent velocity of amyloplast sedimentation in the cytoplasm and gravity variation. The two turning points at 0.25 g and 0.11 g indicate that the displacement of amyloplasts will be much slower under 0.25 g or less gravity. (**b**) At 1*g* gravity, the amyloplasts in the columella cells fall to the lower side of the cell membrane, which is the key step in gravity sensing. The subsequent activation of PIN3 and PIN7 accumulation on the lower side of the cell membrane causes a rapid efflux of auxin. In this case, the auxin content on the upper side of the root tip is higher than the auxin content on the lower side, resulting in rapid growth of the cells in the upper part of the root tip. The root will grow in the direction of gravity. Under the 1/6 g gravity of the Moon, the amyloplasts in the root tip cells are insensitive to gravity and cannot sink to the lower part of the cell membrane rapidly. In this case, PIN3 and PIN7 are evenly distributed in the cell membrane. The auxin on both sides of the root tip is also uniformly distributed, causing the root tip to grow straight. The threshold calculated by the amyloplast precipitation model is 0.25 g. When gravity is less than 0.25 g, the sedimentation rate of the amyloplasts will be greatly reduced, and the root tip cannot bend and grow in response to gravity.

## Data Availability

Not applicable.

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
