# Peer review of "Plant Gravitropism and Signal Conversion under a Stress Environment of Altered Gravity"

_ijms, 2021, doi:10.3390/ijms222111723_

Round 1

Reviewer 1 Report

Overall the review is very interesting and well structured. There are a lot of English language and grammar errors throughout that need to be addressed.

The introductions requires a lot of language and grammar attention as some words are missing from sentences and some words are repeated. Fore example, line 27 'gravity experiments in space biology experiments'. The second 'experiments' can be deleted. Also, line 34 change 'has' to have', line 41 change 'at far side' to 'at the far side'.

Line 30, 'scientists conducted experiments', are you writing this for a newspaper or magazine? Don't use the term 'scientists' in an academic journal. We are all scientists already.

I notice in line 44 you mention to explore other planets, then give the moon as an example. The moon is not a planet. If you are using a different definition of planet, that needs to be given in the text.

Line 56, what does VEGGIE stand for? this needs to be defined in the text.

Line 58 you define what ISS stands for. You have already done this in the introduction, so it is not needed again here.

Line 67, don't use the term 'etc'. Either list and provide examples of everything or pick out a small number of examples. Don't use 'etc'.

Lines 80 to 89 needs language and grammar checking.

Line 91 to 92 is very repetitive of itself. Can this be rewritten?

Line 104, you say 'many experiments' but only provide a single reference. It's either many experiments with several references or it's a single experiment with only one reference. This needs to be amended in the text.

Section 3.2, words are missing in sentences and the flow of the text is hindered. Carefully address the language and grammar in this section.

Section 3.3, you've referred to an equation as a figure then as a formula. They should all be referred to as either a formula or equation.

Line 186 to 187, what does RPM stand for? the text doesn't describe what the M stands for.

Line 200, incorrect referencing format.

Line 232, you provide Figure 2 and it's legend, but Figure 1 isn't labelled and there is no legend for Figure 1.

Line 247, 'amyloplasts fall' are you referring to quantity or are the amyloplasts physically moving?

Line 276, unknown space in outer space' this phrase is a bit clumsy and needs to be rewritten.

Author Response

Response to Referee 1:

Overall the review is very interesting and well structured. There are a lot of English language and grammar errors throughout that need to be addressed.

Answer: Thank you very much for the comments. Our team reviewed the comments one by one carefully, and made the corrections and improvements accordingly. Our native English-speaking colleague have made the corrections too.

The introductions requires a lot of language and grammar attention as some words are missing from sentences and some words are repeated. Fore example, line 27 'gravity experiments in space biology experiments'. The second 'experiments' can be deleted. Also, line 34 change 'has' to have', line 41 change 'at far side' to 'at the far side'.

Answer: Thank you for the suggestions. We made the corrections accordingly. In line 27, we deleted the second “experiments”. In line 34, “has” changed to “have”. In line 41, we added “the”.

Line 30, 'scientists conducted experiments', are you writing this for a newspaper or magazine? Don't use the term 'scientists' in an academic journal. We are all scientists already.

Answer: In line 30-32, we changed the whole sentence into “Plant biology experiments in the ground-controlled environment have contributed to simulate the various changes in plant growth under a space environment of altered gravity”.

I notice in line 44 you mention to explore other planets, then give the moon as an example. The moon is not a planet. If you are using a different definition of planet, that needs to be given in the text.

Answer: In line 44, we changed “further explore other planets (Mars, Moon, etc.)” into “further explore the outer space (Mars, Moon, etc.)”.

Line 56, what does VEGGIE stand for? this needs to be defined in the text.

Answer: In line 56, we changed “NASA launched the ‘VEGGIE Food Production System’” into “ NASA launched the ‘Vegetable Production System (VEGGIE)’”.

Line 58 you define what ISS stands for. You have already done this in the introduction, so it is not needed again here.

Answer: We deleted the “International Space Station”.

Line 67, don't use the term 'etc'. Either list and provide examples of everything or pick out a small number of examples. Don't use 'etc'.

Answer: We deleted the “etc”.

Lines 80 to 89 needs language and grammar checking.

Answer: We made the language and grammar checking in line 80-89. The corrections and improvements were made accordingly. Thank you very much for the suggestions.

Line 91 to 92 is very repetitive of itself. Can this be rewritten?

Answer: Yes, line 91 to 92 were rewritten as “Gravity sensing in plants is a very complex process, including the perception of gravity by cells and signal transduction in cells, as well as the cells response to change.”

Line 104, you say 'many experiments' but only provide a single reference. It's either many experiments with several references or it's a single experiment with only one reference. This needs to be amended in the text.

Answer: We added more references in line 104 for 'many experiments'. Thank you for the suggestion.

Section 3.2, words are missing in sentences and the flow of the text is hindered. Carefully address the language and grammar in this section.

Answer: Thank you very much for the comments. We carefully checked all the language and grammar in this section, and made the corrections accordingly.

Section 3.3, you've referred to an equation as a figure then as a formula. They should all be referred to as either a formula or equation.

Answer: Yes, we made the change, to assign the reference as either a formula or equation. Thank you for the suggestion.

Line 186 to 187, what does RPM stand for? the text doesn't describe what the M stands for.

Answer: We change “Random positioners (RPMs)” into “Random Positioning Machine (RPM)” in line 186-187.

Line 200, incorrect referencing format.

Answer: We deleted “Manzano et al reported that” and added the reference number in the end of sentence.

Line 232, you provide Figure 2 and it's legend, but Figure 1 isn't labelled and there is no legend for Figure 1.

Answer: We added the legend for Figure 1 in line 164-165.

Line 247, 'amyloplasts fall' are you referring to quantity or are the amyloplasts physically moving?

Answer: It is amyloplasts physically moving.

Line 276, unknown space in outer space' this phrase is a bit clumsy and needs to be rewritten.

Answer: We deleted “unknown space in”. Thank you very much again for all the comments. The quality of manuscript was significantly improved following these suggestions.

Reviewer 2 Report

I would suggest to describe in more detail the experiment conducted on the moon. How may seeds were used? Are they all belonging to a uniform variety or representing a segregating population? This is relevant to understand if the there a genetic variability for low gravity response or not. If they belong to a single variety, which one?

I do not understand the reference to rapeseed in the caption of Fig. 2 at page 6. Was the the experiment carried out on cotton or rapeseed?

Author Response

Response to Referee 2:

I would suggest to describe in more detail the experiment conducted on the moon. How may seeds were used? Are they all belonging to a uniform variety or representing a segregating population? This is relevant to understand if the there a genetic variability for low gravity response or not. If they belong to a single variety, which one?

Answer: Thank you very much for the comments. Our team reviewed the comments one by one carefully, and made the corrections and improvements accordingly. We did provided more details for this experiments: “The biological experiment payload (BEP) is a cylindrical structure with a weight of 2.608 kg, which meets the requirements of the Chang’e 4 probe. The BEP includes a control module, a thermal control module, a structural module, a light-guiding module, and a biological module. The biological space has a volume of approximately 0.82 L, contains cotton seeds, rapeseed seeds, potato seeds, Arabidopsis seeds, insects and yeast, and constitutes a micro-life cycling system. The BEP has a very small light-transmitting hole with a diameter of Φ10.” Please see section 4.2 for more details.

The BEP carried four cotton seeds, ten rapeseed seeds, ten potato seeds and ten Arabidopsis seeds. All the seeds for each species are belonging to a uniform variety since the experiments were designed to study the plant response under the real moon environment with strong radiation, micro gravity and hot temperature. Those varieties were bred by our team with few years of selections under the simulated stress environment. Thank you again for the suggestions.

I do not understand the reference to rapeseed in the caption of Fig. 2 at page 6. Was the the experiment carried out on cotton or rapeseed?

Answer: Thank you for the comments. The experiment was carried out with cotton seeds, rapeseed seeds, potato seeds and Arabidopsis seeds. We added more details in the section 4.2.
